# Similarities and Differences between Vestibular Migraine and Recurrent Vestibular Symptoms—Not Otherwise Specified (RVS-NOS)

Roberto Teggi [1,*], Bruno Colombo [2], Iacopo Cangiano [1], Omar Gatti [1], Mario Bussi [1] and Massimo Filippi [2,3]

1    Department of Otolaryngology, IRCCS San Raffaele Scientific Institute, Via Olgettina 60, 20132 Milan, Italy; cangiano.iacopo@hsr.it (I.C.); gatti.omar@hsr.it (O.G.); bussi.mario@hsr.it (M.B.)
2    Neurology Unit, IRCCS San Raffaele Scientific Institute, 20132 Milan, Italy; colombo.bruno@hsr.it (B.C.); filippi.massimo@hsr.it (M.F.)
3    Faculty of Medicine, Vita-Salute San Raffaele University, 20132 Milan, Italy
*    Correspondence: teggi.roberto@hsr.it

**Abstract:** Menière's disease and vestibular migraine (VM) are two common inner ear disorders whose diagnoses are based on clinical history and audiometric exams. In some cases, patients have been reporting different episodes of vertigo for years but not fulfilling the Bárány Society criteria for either. These are called Recurrent Vestibular Symptoms—Not Otherwise Specified (RVS-NOS). It is still under debate if this is a single disease entity or a part of the spectrum of already established disorders. The purpose of our work was to establish similarities and differences with VM in terms of clinical history, bedside examination, and family history. We enrolled 28 patients with RVS-NOS who were followed for at least 3 years with stable diagnosis; results were compared with those of 34 subjects having a diagnosis of definite VM. The age of onset of vertigo was lower in VM than in RVS-NOS (31.2 vs. 38.4 years). As for the duration of attacks and symptoms, we detected no differences other than subjects with RVS-NOS reporting milder attacks. Cochlear accompanying symptoms were more frequently reported by VM subjects (one subject reporting tinnitus and another one reported tinnitus and fullness). Motion sickness was equally reported by subjects across two samples (around 50% for both). Bipositional long-lasting, non-paroxysmal nystagmus was the most common finding in the two groups, with no significant difference. Finally, the percentage of familial cases of migrainous headache and episodic vertigo did not differ between the two samples. In conclusion, RVS-NOS shares some common aspects with VM, including the temporal profile of attacks, motion sickness (commonly considered a migraine precursor), bedside examination, and family history. Our results are not inconsistent with the possibility that RVS-NOS may be a heterogeneous disorder, even if some of these subjects may share common pathophysiological mechanisms with VM.

**Keywords:** vertigo; vestibular migraine; RVS-NOS; vestibular bedside; motion sickness

## 1. Introduction

The two most common vestibular disorders provoking attacks lasting more than 5 min are vestibular migraine (VM) and Menière's disease (MD). The Bárány Society has proposed diagnostic criteria for both [1,2]. The presence of migrainous headache and cochlear symptoms is crucial for differential diagnosis between the two forms.

In some cases, the patient reports episodic vertigo without concomitant cochlear symptoms and does not present lifetime migrainous headache or photo-phonophobia concomitant to vertigo, and thus does not fit a diagnosis of VM or MD according to the Bárány Society criteria. This form has been previously called benign recurrent vertigo or recurrent vestibulopathy [3,4], but both terms go back to years in which diagnostic criteria for VM and MD had not been published.

Previous works described heterogeneous attacks lasting from 1 min to 24 h, from mild to severe intensity, while other authors reported concomitant cochlear symptoms or headaches in some cases [5–7].

More recently, similarities and differences between RVS-NOS and MD and VM were investigated; it was concluded that the duration of attacks and clinical features share some similitudes with VM rather than MD [8].

The purpose of our work was to collect clinical characteristics, bedside examination, and family history in subjects with RVS-NOS and compare these with a sample of patients with VM to establish possible similarities and differences.

## 2. Materials and Methods

### 2.1. Study Design and Participants

In this retrospective study, 28 patients diagnosed with RVS-NOS were enrolled. Data were retrieved from clinical records. Patients were enrolled in our Center for Vestibular Disorders and included if they had at least five episodes of vertigo lasting more than 5 min. RVS-NOS was defined as an episodic vestibular syndrome with at least five episodes of vestibular symptoms according to the Bárány Vestibular Symptoms grid [9] that could not better be explained by another vestibular disorder, with each episode of vertigo lasting more than 5 min.

The inclusion criteria were:

a. A negative lifetime history of migrainous headaches.
b. Absence of photo/phonophobia during vertigo and fluctuating hearing loss even in a vertigo-free period.
c. Audiometric exam showing normal hearing level or moderate hearing loss on acute tones, symmetric on both sides.
d. No residual instability after the resolution of vertigo attacks.
e. A negative central nervous system MRI other than for microischemic lesions.
f. A 3-year follow-up with stable concomitant symptoms.

The following clinical data were collected by a senior neurotologist:

a. Age of onset of vertigo.
b. Duration of vertigo attacks; for this point, multiple responses were possible. Regarding duration, we saved data in five categories: less than 1 h, from 1 to 6 h, from 6 to 12 h, from 12 to 24 h, and more than 1 day.
c. Nausea and vomiting during vertigo; the presence of cochlear symptoms, hearing loss, tinnitus, and fullness, at least in some cases.
d. Family history of vertigo and headache. Patients were also asked if any of their family members, i.e., another relative in the first or second degree, suffered from migraine and/or vertigo.
e. Concomitant cochlear symptoms, aural fullness, and tinnitus, in at least one episode.

According to the Bárány Society criteria [9], patients were asked to characterize vestibular symptoms by choosing among (multiple answers were allowed):

a. Internal vertigo is defined as the false sensation of self-motion (spinning, swaying, tilting, bobbing, bouncing), when no self-motion is occurring.
b. Dizziness is the sensation of disturbed or impaired spatial orientation without a false or distorted sense of motion.
c. Visuo-vestibular symptoms-external vertigo is the false sensation that the visual surrounding is spinning or flowing.

Finally, patients were asked if most vertigo attacks (more than 50%) were of severe intensity according to the Bárány Society criteria [9]. It was considered severe when daily activities were not possible.

*2.2. Procedures and Measurements*

A bedside examination was performed by a senior neurotologist with infrared video goggles (Interacoustics, Taastrup, Denmark). Spontaneous nystagmus was assessed with the patient seated in a clinical chair, in the primary position and eyes rotated 15° on the right and left side, while positional nystagmus was assessed in the supine position with the head turned 90° on both sides.

Video head impulse test was performed with a commercially available system (ICS Impulse, Otometrics, Taastrup, Denmark) on the horizontal plane. Trials with blinks and outliers were automatically excluded. Patients in whom recordings demonstrated that eye movements preceded head movements, even after attempts to improve goggle fit, were not included. The test was considered negative when the VOR gain was over 0.8 and when no corrective saccades were detected.

Head-shaking test with the patient sitting in a clinical chair with the head tilted downwards by 30°. The patient's head was vigorously rotated 20 times on the horizontal plane with a maximum amplitude of 30–40°. Post-HST nystagmus was recorded for 1 min and was considered positive when nystagmus lasting at least 5 s was detected.

A skull vibration test (skull vibration-induced nystagmus—SVIN test) was performed at 100 Hz with a commercially available system (VVIB—Synapsis). Stimuli were applied perpendicularly to the skin over the mastoid process, posterior to the auricle, at the level of the external acoustic meatus and on the midline with a force of around 1 kg; three stimulation trials were performed on each mastoid, lasting 5–10 s each. Eye movements were studied with video Frenzel goggles and visual fixation of both eyes was inhibited. The test was considered positive when horizontal nystagmus, always beating on the same side, was elicited in all six trials [10].

Finally, Dix Hallpike and Pagnini McClure maneuvers were performed to exclude BPPV.

Results of clinical data and bedside examination were compared with those of 34 patients diagnosed with definite VM according to Bárány Society criteria [11]. They were enrolled between January and April 2023 in our Center for Vestibular Disorders and were never included in previous studies.

We obtained the approval of our Ethics Committee for a larger study on VM and episodic vertigo on 11 December 2014.

*2.3. Statistical Analysis*

Absolute and relative frequencies of each symptom were calculated and compared between groups using chi-square statistics. Quantitative variables are presented as mean ± standard deviation. A *p* value < 0.05 was considered statistically significant. Analyses were carried out with SPSS (version 22).

**3. Results**

The age of onset of vertigo in VM subjects was 31.2 ± 9.6 years compared to 38.4 ± 8.4 years in those with RVS-NOS (t = 3.1, *p* = 0.003). Twenty-seven (79%) subjects with VM were females vs. fifteen (54%) with RVS-NOS (*p* = 0.03). The disease duration was 5.8 ± 1.5 years in RVS-NOS and 8.1 ± 2 years in the VM sample.

An overlapping profile of vertigo attacks was reported in the different groups. In the VM sample, 12 patients (35.7%) reported external vertigo, 13 (38.2%) internal vertigo, and 15 (44.1%) dizziness compared to 8 (28.6%), 11 (39.2%), and 16 (57.1%) in RVS-NOS subjects, respectively ($\chi^2$ = 0.6, *p* = 0.7). Severe attacks were more frequently reported in most patients with VM (20/38, 58.8%) compared to those with RVS-NOS (8/20, 28.6%; t = 5.7, *p* = 0.02).

Nausea was reported in most attacks in 31 patients (92.1%) with VM, while 24 (85.7%) subjects with RVS-NOS (*p* = 0.5); 12 patients (35.3%) with VM reported vomiting in most attacks vs. 3 in RVS-NOS ($\chi^2$ = 5, *p* = 0.02).

Results of the duration of vertigo attacks are reported in Table 1.

**Table 1.** Duration of vertigo attacks in VM and RVS-NOS. Multiple responses were possible. No significant difference was detected between the two groups ($\chi^2 = 1.7$, $p = 0.7$).

|  | VM (n = 34) | RVS-NOS (n = 28) |
|---|---|---|
| LESS THAN 1 h | 6 | 8 |
| 1–6 h | 14 | 13 |
| 6–12 h | 11 | 7 |
| 12–24 h | 6 | 3 |
| MORE THAN 1 DAY | 1 | 1 |

Tinnitus during vertigo was reported in at least one episode by ten (29.4%) of those with VM and in two patients (7.1%) with RVS-NOS ($\chi^2 = 4.8$, $p = 0.03$); other cochlear symptoms, hearing loss, or fullness, were reported by nine (26.5%) patients with VM and only one (3.5%) with RVS-NOS ($\chi^2 = 5.9$, $p = 0.01$). Cochlear symptoms in the VM group were reported by sixteen patients: three had all three symptoms, six had fullness without tinnitus, and seven had only tinnitus. Among patients with RVS-NOS, one reported fullness and tinnitus, and another only tinnitus.

A similar percentage of subjects with motion sickness in a pediatric age was seen in the two samples: 14 (65.0%) in VM and 13 (46.4%) in RVS-NOS.

Only five subjects (three in the VM sample) exhibited mild symmetric hearing loss on acute frequencies; all other subjects had a normal hearing level.

At bedside examination, none of the patients in the two groups had a positive video head impulse test. The most common finding in both samples was bipositional, long-lasting (more than 1 min), non-paroxysmal, apogeotropic nystagmus, which was present in eight (23.5%) patients with VM and four (14.3%) with RVS-NOS ($\chi^2 = 1.2$, $p = 0.2$). A nystagmus was provoked by head shaking or skull vibration test in four (10.5%) patients with VM and in two with RVS-NOS (7.1%).

No significant difference was detected between the two samples of family history for headache, reported by 20 (58.8%) patients with VM and 11 (39.3%) with RVS-NOS ($\chi^2 = 2.3$, $p = 0.1$), or vertigo, respectively reported by 10 (29.4%) and 4 (14.3%) patients in the two groups ($\chi^2 = 2$, $p = 0.2$).

Lastly, 12 subjects (42.8%) with RVS-NOS reported headaches without clinical features for migraine according to ICHD criteria [12]; all of them had a diagnosis of tension-type headaches made by a senior neurologist.

Clinical data are summarized in Table 2.

**Table 2.** Results of clinical history, symptoms, accompanying symptoms, bedside examination, and family history in the VM and RVS-NOS groups. Total values and percentage (between parentheses) *p* values are reported in the last column; n.s. = not significant.

|  | VM (n = 34) | RVS-NOS (n = 28) | *p* |
|---|---|---|---|
| Female | 27 (79%) | 15 (54%) | 0.03 |
| Age of onset of vertigo (y) | 31.2 ± 9.6 | 38.4 ± 8.4 | 0.003 |
| Symptoms |  |  |  |
| External vertigo | 12 (35.3%) | 8 (28.6%) | n.s. |
| Internal vertigo | 13 (38.2%) | 11 (39.2%) | n.s. |
| Dizziness | 15 (44.1%) | 16 (57.1%) | n.s. |
| Severe attacks | 20 (58.8%) | 8 (28.6%) | 0.002 |
| Nausea | 31 (92.1%) | 24 (85.7%) | n.s. |
| Vomiting | 12 (35.3%) | 3 (10.7%) | 0.02 |
| Headache of any kind | 34 (100%) | 12 (42.8%) | ≤0.00001 |
| Tinnitus | 10 (29.4%) | 2 (7.1%) | 0.02 |
| Fullness/hearing loss | 9 (26.5%) | 1 (3.5%) | 0.004 |
| Motion sickness | 17 (50%) | 13 (46.4%) | n.s. |
| Bedside |  |  |  |
| SVINT/hst | 3 (8.8%) | 2 (7.1%) | n.s. |
| Positional nystagmus | 8 (23.5%) | 4 (14.3%) | n.s. |
| Family history |  |  |  |
| Headache | 20 (58.8%) | 11 (39.3%) | n.s. |
| Vertigo | 10 (29.4%) | 4 (14.3%) | n.s. |

## 4. Discussion

The nature of RVS-NOS is still under debate, as well as if the condition may share pathophysiological mechanisms with VM. Studies before the publication of diagnostic criteria for VM often did not exclude patients with migrainous headaches and/or photo-phonophobia during vertigo; these patients would now be diagnosed with probable VM [13,14]. Our sample was composed of patients with a negative lifetime history of migrainous headache and they did not report photo or phonophobia during vertigo, thus excluding diagnoses of probable\definite VM according to the Bárány Society criteria.

In our study, we wanted to evaluate the similarities and differences between VM and RVS-NOS. Heterogeneous results have been reported in the few studies on these subjects, some of which underlined possible overlaps with migraine [14,15]. We applied strict inclusion criteria for RVS-NOS to avoid overlapping with probable VM and with symptoms and diagnoses that were stable over time.

Firstly, unlike patients with VM samples, those with RVS-NOS present an equal sex distribution. Our results are in line with those reported in recent publications [5,8], while other authors, not unlike VM, reported a female predominance [16,17]. The different inclusion criteria can likely explain this, and a possible bias may be the inclusion of subjects with probable VM in RVS-NOS in some studies. In support of this possibility, a previous investigation reported a female preponderance only in patients with benign recurrent vertigo with migraine [7].

We found a lower age of onset of vertigo in VM, which was $38.4 \pm 8.4$ years in RVS-NOS. Previous authors reported the age of the first attack ranged between 31 and 55 years [5,8,16–18].

The characteristics of vertigo overlap in the two groups, with no difference for external/internal vertigo and dizziness; moreover, no difference was detected for the duration of attacks, which was highly variable in both groups. On the other hand, in line with other reports, patients with RVS-NOS reported severe attacks less frequently compared to those with VM [5,8].

Concomitant cochlear symptoms were less frequent in RVS-NOS, reported in only two subjects, although previous studies reported overlapping data for VM subjects [19,20]. Only one recent study reported cochlear symptoms, which were present in 51% of patients [5]. Of note, considering a possible overlap between VM and MD, an audiometric exam in all subjects ruled out a possible diagnosis of MD [21].

The most common finding at bedside examination in the two groups was a bipositional long-lasting, low frequency, non-paroxysmal, apogeotropic nystagmus. It should be noted that positional nystagmus with these features is probably the most frequent clinical sign in VM when patients are evaluated outside vertigo attacks and is considered a sign of a central vestibular disorder [22,23]. As far as we know, only one other study has reported data on positional nystagmus in RVS-NOS, which was found in 17% of subjects with benign recurrent vertigo [17].

In our opinion, interesting data can be drawn from the percentage of patients reporting motion sickness and a family history of migraine and episodic vertigo.

Motion sickness is equally reported in patients with RVS-NOS and VM, being 50% in both. Our data overlap with those reported in a previous investigation [17]. Although motion sickness is a common finding in the general population and has a strong genetic contribution, it is commonly considered to be a migraine precursor [24,25].

Lastly, 58.8% of patients with VM reported that a family member of first-second degree suffered from migrainous headache versus 39.3% of those with RVS-NOS, while 29.4% and 14.3%, respectively, referred to familial episodic vertigo. We found a slightly lower proportion of family migraine than in a previous study, reported in 56% of those with RVS-NOS [5]. Another investigation reported familial cases of dizziness in 23% of subjects [17]. It should be noted that diagnoses were referred by patients and could not be confirmed by clinical examination. It could be speculated that the non-significant difference

in familial cases for both migraine and vertigo may underline some possible common pathophysiological mechanisms between VM and RVS-NOS.

This study has some limitations. First, the small sample of RVS-NOS and the inclusion of VM subjects recruited in a limited period of time should be considered, since we wanted to avoid including patients that had been included in previous studies. Moreover, since data on RVS-NOS were retrospectively retrieved from our records, all patients only performed routine screening with a limited number of tests; further studies should assess vestibular function through additional tests like caloric tests or VEMPs to detect potential differences between VM and RVS-NOS.

## 5. Conclusions

Additional studies are required to understand if RVS-NOS is a poorly known single disorder or a heterogeneous one. Of note, there are no diagnostic criteria, and previous investigations reported on non-homogeneous cohorts. The present study assessed common features between RVS-NOS and VM. The temporal profile of the duration of attacks is similar, although those with RVS-NOS frequently reported mild attacks.

Moreover, motion sickness in pediatric age and commonly considered a migraine precursor, was reported by similar proportions of patients in the two groups.

Family history for migraine and vertigo was somewhat less frequent in those with RVS-NOS, although the difference was not statistically significant. Although not conclusive, the present results are not inconsistent with the possibility that a proportion of patients with RVS-NOS might share common genetic and pathophysiological mechanisms with VM.

**Author Contributions:** Conceptualization and methodology, B.C. and R.T.; data acquisition, I.C. and O.G.; data analysis and interpretation R.T.; original draft preparation, I.C. and R.T.; writing-review and editing, R.T.; supervision, M.B. and M.F. All authors made substantive intellectual contributions to the manuscript, revised it, and provided substantial comments. All authors have read and agreed to the published version of the manuscript.

**Funding:** This study received no external funding.

**Institutional Review Board Statement:** This study adheres to the principles of the Helsinki Declaration and was approved in 2014 by the Ethics Committee of our hospital.

**Informed Consent Statement:** Since data were retrieved from our records, no informed consent was signed. All patients only performed routine screening.

**Data Availability Statement:** The data supporting the presented results of the study are available on request from the corresponding author. The data are not publicly available to protect the privacy of the participants.

**Conflicts of Interest:** The authors declare no conflict of interest.

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
