# Peer review of "Similarities and Differences between Vestibular Migraine and Recurrent Vestibular Symptoms—Not Otherwise Specified (RVS-NOS)"

_audiolres, doi:10.3390/audiolres13030041_

Round 1
Reviewer 1 Report
Congratulation to the well written papers pertaining to the clear and rigorous classification of both studied conditions. This makes the study very importnatn from the methodological point.
I have some following remarks to consider:
Could you specify the disease duration in your groups? It would be important, if there was possible to observe transition from RVS-NOS to VM or vice versa in some cases.
It would enhance the clarity of the paper, if you will specify the features missing in RVS-NOS, which prevent their classification as VM.
Have you followed the response to antimigrainous therapy in both groups and possible differences?
please note the paragraph on the rows 121-125 - there seems to be a misconception – it starts with claim, that all patinets fulfilled criteria for VM and ends with annother claim, that all patients fulfilled criteria for RVS-NOS
I have found some typing errors:
row number:
10 ….vestibilar migraine… should be „vestibular“
33 tha… should be „than“
107 not „for 20 times“ it should be „20 times“
189 recently publications --- should be „recent publications“
190 it is likely, that the different inclusion criteria – for better comprehension the word „ that“ is missing
188 smples should be „samples“
200 consider reformulation eg: „in line with other reports“
257 should be „data retrieved from our records“ (not „by our records“)
258 All patients only performed routine screening. lépe :
Results of standard neurotological clinical examination were available in all included patients.
261 fullstop is missing at the end of phrase
Author Response
Congratulation to the well written papers pertaining to the clear and rigorous classification of both studied conditions. This makes the study very importnatn from the methodological point.
Thank you for your considerations; we included all suggestions.
I have some following remarks to consider:
Could you specify the disease duration in your groups? It would be important, if there was possible to observe transition from RVS-NOS to VM or vice versa in some cases.
We added the duration. It should be considered that RVS-NOS patients were selected in our records if having episodic vertigo for at least 3 years and still not having criteria to be diagnosed with VM
It would enhance the clarity of the paper, if you will specify the features missing in RVS-NOS, which prevent their classification as VM.
A sentence has been added in the discussion
Have you followed the response to antimigrainous therapy in both groups and possible differences?
We have data of therapy with calcium channel blockers in these subjects, some of them were responders; we tried to give an answer to the question, but statistics would be probably more confounding
please note the paragraph on the rows 121-125 - there seems to be a misconception – it starts with claim, that all patinets fulfilled criteria for VM and ends with annother claim, that all patients fulfilled criteria for RVS-NOS
The final misleading sentence has been removed.
Comments on the Quality of English Language
I have found some typing errors:
row number:
10 ….vestibilar migraine… should be „vestibular“
33 tha… should be „than“
107 not „for 20 times“ it should be „20 times“
189 recently publications --- should be „recent publications“
190 it is likely, that the different inclusion criteria – for better comprehension the word „ that“ is missing
188 smples should be „samples“
200 consider reformulation eg: „in line with other reports“
257 should be „data retrieved from our records“ (not „by our records“)
258 All patients only performed routine screening. lépe :
Results of standard neurotological clinical examination were available in all included patients.
261 fullstop is missing at the end of phrase
Typing errors have been corrected

Reviewer 2 Report
This is a very interesting study that compares the clinical findings that differentiate, or are common, between two relatively frequent entities in the otoneurology clinic: MD vs. SVR-NOS. Although the results are interesting and there seems to be some similarity between the two clinical pictures, I would like the authors to complete some suggestions and doubts that arise while reading them.
In Abstract, it says: "The two most common episodic vertigos are Meniere's disease and vestibular migraine (VM)", of course, vestibular should be changed to vestibular, but also this sentence is not exactly adequate, since the most frequent vestibular recurring condition is the BPPV. This could be worded differently.
In Methods, the authors must include the data of the Ethics Committee that authorized it.
On the other hand, both groups studied presented cochlear symptoms in the form of hearing loss, etc. It would be interesting to know the results of the audiometry in both groups studied.
It is evident that the RVS-NOS group presented a highly significant association with non-migraine headache (42%) and probably less pain intensity (if they were compared with a control group). But, would it be possible to know the diagnosis or type of headache in this study group?
In Discussion: Other studies have shown that in some SVR-NOS patients with accompanying ear symptoms, attack durations of <20 min excluded them from the diagnosis of MD. In this sense, the study of the inflammatory profile of the blood could help to distinguish between MD vs SVR-NOS, specifically a study identified the levels of IL-1β, CCL3, CCL22 and CXCL1 as capable of differentiating patients with MV from patients with MD with a high degree of accuracy, suggesting a powerful diagnostic value in patients with overlapping symptoms (reference):
-Flook M, Frejo L, Gallego-Martinez A, Martin-Sanz E, Rossi-Izquierdo M, Amor-Dorado JC, Soto -Varela A June 4, 2019; 10: 1229. doi: 10.3389/fimmu.2019.01229. PMID: 31214186; PMCID: PMC6558181).
Authors should consider including some reference to it.
Review some wording mistake
Author Response
Comments and Suggestions for Authors
This is a very interesting study that compares the clinical findings that differentiate, or are common, between two relatively frequent entities in the otoneurology clinic: MD vs. SVR-NOS. Although the results are interesting and there seems to be some similarity between the two clinical pictures, I would like the authors to complete some suggestions and doubts that arise while reading them.
Thank you for your comments; we included in the revision your suggestions.
In Abstract, it says: "The two most common episodic vertigos are Meniere's disease and vestibular migraine (VM)", of course, vestibular should be changed to vestibular, but also this sentence is not exactly adequate, since the most frequent vestibular recurring condition is the BPPV. This could be worded differently.
The sentence has been rewritten
In Methods, the authors must include the data of the Ethics Committee that authorized it.
It has been added
On the other hand, both groups studied presented cochlear symptoms in the form of hearing loss, etc. It would be interesting to know the results of the audiometry in both groups studied.
A sentence has been added
It is evident that the RVS-NOS group presented a highly significant association with non-migraine headache (42%) and probably less pain intensity (if they were compared with a control group). But, would it be possible to know the diagnosis or type of headache in this study group?
A sentence has been added.
In Discussion: Other studies have shown that in some SVR-NOS patients with accompanying ear symptoms, attack durations of <20 min excluded them from the diagnosis of MD. In this sense, the study of the inflammatory profile of the blood could help to distinguish between MD vs SVR-NOS, specifically a study identified the levels of IL-1β, CCL3, CCL22 and CXCL1 as capable of differentiating patients with MV from patients with MD with a high degree of accuracy, suggesting a powerful diagnostic value in patients with overlapping symptoms (reference):
-Flook M, Frejo L, Gallego-Martinez A, Martin-Sanz E, Rossi-Izquierdo M, Amor-Dorado JC, Soto -Varela A June 4, 2019; 10: 1229. doi: 10.3389/fimmu.2019.01229. PMID: 31214186; PMCID: PMC6558181).
Authors should consider including some reference to it.
A sentence has been added and reference included
Comments on the Quality of English Language
Review some wording mistake
